# Oligomeric Proanthocyanidins Alleviate the Detrimental Effects of Dietary Histamine on Intestinal Health of Juvenile American Eels (*Anguilla rostrata*)

**Shuo Wang [1], Yingxia He [1], Feng Xi [1], Ying Liang [1,2] and Shaowei Zhai [1,2,*]**

[1]  Engineering Research Center of Modern Industry Technology for Eel, Ministry of Education, Fisheries College of Jimei University, Xiamen 361021, China

[2]  Key Laboratory of Healthy Mariculture for the East China Sea, Ministry of Agriculture and Rural Affairs, Jimei University, Xiamen 361021, China

*  Correspondence: zhaisw@jmu.edu.cn; Tel.: +86-592-6181420

**Abstract:** This study was conducted to evaluate the oligomeric proanthocyanidins (OPC) in alleviating the detrimental effects of intestinal health caused by dietary histamine in juvenile American eels (*Anguilla rostrata*). A total of 480 fish with a similar body weight of $10.84 \pm 0.16$ g were randomly divided into four groups, and there were the control group fed a basal diet, the HIS group fed a diet with a high level of histamine (534 mg/kg), the H + OPC I group fed the high histamine diet with 300 mg/kg OPC, and the H + OPC II group fed the high histamine diet with 600 mg/kg OPC, respectively. After the fish were fed the trial diets for 77 days, the intestinal samples were taken, and the related parameters of intestinal health were analyzed. Dietary 300 mg/kg or 600 mg/kg OPC could reverse the decreased activities of lipase, protease, and glutathione peroxidase and the level of total antioxidant capacity, the increased intestinal malondialdehyde and D-lactate acid levels and the activity of diamine oxidase in serum, and the decreased villus height caused by a high level of dietary histamine. There were no significant differences above all the indices between the H + OPC I group and the H + OPC II group. The higher relative abundances of potentially pathogenic bacteria were induced by the high level of dietary histamine. Dietary 300 mg/kg OPC might increase the relative abundance of the potential probiotics and inhibit the colonization of intestinal pathogenic bacteria of juvenile American eels exposed to the stress of high dietary hisatamine. The intestinal health status of the H + OPC groups was similar to that of the control group. These results suggested that dietary 300 mg/kg OPC might alleviate the detrimental effects of dietary 534 mg/kg histamine on the intestine health of juvenile American eels by increasing the activity of digestive enzymes, improving the antioxidative potential and barrier function, and beneficially modulating the intestinal microbiota. Dietary 600 mg/kg OPC could not exert further improvement in growth performance and the intestinal health of juvenile American eels.

**Keywords:** oligomeric proanthocyanidins; histamine; intestinal health; *Anguilla rostrata*

**Key Contribution:** Dietary supplementation of oligomeric proanthocyanidins extract from grape seed might alleviate the detrimental effects of high level of dietary histamine on intestine health in juvenile American eels by increasing the digestive enzymes activity, strengthening the barrier function and antioxidative potential in the intestine, and beneficially shaping the composition of intestinal microbiota.

## 1. Introduction

At present, the American eel (*Anguilla rostrata*) is the major eel species extensively cultivated in the Fujian Province of China [1]. As a typical carnivorous fish, it has a high standard for protein requirement in the diet. The white fish meal is generally provided as the main ingredient of the eel diet, which accounts for 60–70% [2]. To alleviate the pressure

of the rising price of white fish meal, the proportion of brown fish meal with lower price is constantly increasing in the eel diet in recent years. It is well-known that the histamine level in the white fish meal is much lower than that of the brown fish meal. Histamine is the most toxic biogenic amine in fish meal, which is formed by the decarboxylation of L-histidine during fish meal manufacture, transportation, and storage [3,4]. It was found that excessive histamine in the diet could cause oxidative stress by promoting the production of reactive oxygen species (ROS), which might be the major factor in inducing the damage effects on intestinal health and ultimately decreasing the growth performance of eels [5]. The problems in intestinal health caused by dietary histamine have attracted widespread attention in eel culture in recent years. In the field of practical aquaculture, functional feed additives with antioxidant potential are usually used to alleviate the oxidative stress caused by different sources of stressors [6–8].

Among those functional feed additives, natural plant polyphenols demonstrate excellent ability to subdue oxidative stress. Oligomeric proanthocyanidins (OPC), as one of the proanthocyanidins, is composed of two monomers, catechin, and epicatechin, in oligomeric or polymeric form, and they are widely distributed in plants [9,10]. OPC possesses strong antioxidant activity and other biological activities, including antibacterial, anti-inflammatory, antiallergic, antiviral, etc. [6,11,12]. OPC has been widely applied in the field of the food industry, pharmaceutical industry, and feed industry [12–16]. For its application in the feed of aquatic animals, the protection effects of dietary OPC supplementation were reported to increase antioxidant potential in the intestine and alleviate the damage of intestinal health caused by dietary cadmium stress in tilapia (*Oreochromis niloticus*) [8] and pearl gentian grouper (♀*Epinephelus fuscoguttatus* × ♂*Epinephelus lanceolatu*) [17]. A similar effect was also found in greenlip abalone (*Haliotis laevigata*) under high-temperature stress [18]. In a recent study, OPC could counteract growth inhibition by beneficially regulating the serum biochemical indexes and liver metabolites of American eel juveniles exposed to a high level of dietary histamine [7]. The changes in intestinal health of this fish species were not explored, which might be relevant to the beneficial effect of growth performance and liver metabolism. Therefore, the current research was conducted to investigate the effects of dietary OPC supplementation on intestinal health parameters including digestive enzymes, antioxidant potential, intestinal barrier function, and intestinal microbiota of American eels exposed to a high level of dietary histamine.

## 2. Materials and Methods

### 2.1. Experimental Fish, Diet, and Management

After the adaptation period of 4 weeks, 480 juvenile American eels were divided into four groups randomly, and there were the control group fed the basal diet, the HIS group fed a diet with a high level of histamine, the H + OPC I group fed a diet with a high level of histamine and 300 mg/kg OPC, and a H + OPC II group fed a diet with a high level of histamine and 600 mg/kg OPC, respectively. There were four replicates with each group, and each tank contained 30 fish. The feeding trial lasted for 77 days.

Because the samples for measuring the intestinal parameters of juvenile American eels in the present study were taken from our previous feeding trial, the fish acclimation, ingredients and preparation of the diets, OPC source, feeding practice, water quality maintenance, and other trial fish management were consistent with those of the feeding trial in our previous research [7]. The relevant description was not presented thoroughly in this trial. The histamine level of the basal diet was 217 mg/kg, and the histamine level in the diet of the HIS group was 534 mg/kg, which was measured using a colorimetric enzyme method, and the assay kit was provided by the Kikkoman Company in Japan.

### 2.2. Sample Collection

After the last feeding of the trial, eels were fasted for 24 h prior to intestinal sample collection. Nine eels per tank were randomly chosen to be anesthetized with 100 mg/L of eugenol and tail vein blood was sampled for analysis of serum parameters according

to the method of Xu et al. [12]. The intestinal tissues were dissected with three eels per tank and submerged in Bouin's solution for the measure and analysis of intestinal morphology. Following that, the intestines (1–2 g) of three eels from each tank were mixed and homogenized with the precooled normal saline (0.86%) of 10 times volume. The mixture was ground to obtain the homogenate and centrifuged at 3000 r/min for 10 min at 4 °C, and the supernatant was sampled for subsequent analysis of the digestive enzyme activities and the parameters related to antioxidant potential. The foregut samples were collected from three eels per tank and stored at −80 °C for analyzing the intestinal microbiota.

### 2.3. Intestinal Digestive Enzyme Analysis

The intestinal protease activity was determined with the Folin-phenol reagent method. Activities of intestinal amylase and lipase were measured using commercial kits (Nanjing Jiancheng Bioengineering Institute, Nanjing, China).

### 2.4. Intestinal Antioxidant Potential

The malondialdehyde (MDA), total antioxidant capacity (T-AOC), catalase (CAT), superoxide dismutase (SOD), and glutathione peroxidase (GSH-PX) were determined using commercial kits methods (Nanjing Jiancheng Bioengineering Institute, Nanjing, China).

### 2.5. Parameters Related to Intestinal Barrier Function

The commercial kits for the measurement of D-lactate (D-lac) level and diamine oxidase (DAO) activity in the serum were provided by Nanjing Jiancheng Bioengineering Institute (Nanjing, China). The sections of intestine tissue were manufactured according to the methods described by Chen et al. [19].

### 2.6. Intestinal Microbiota Analysis

The intestinal microbiota analysis and processing were referred to the previous methods in the study of Shi et al. [20]. The high throughput sequencing of the amplified 16S rRNA V3-V4 region of intestinal DNA samples was performed using the Illumina Miseq PE300 high throughput sequencing platform in Beijing Allwegen Gene Technology Co., Ltd. (Beijing, China).

### 2.7. Statistical Analysis

The differences in digestive enzymes activities, antioxidant potential, and parameters related to intestinal barrier function among the different groups were followed by one-way analysis of variance (ANOVA) by the SPSS 20.0 (SPSS Inc., Chicago, IL, USA), and the data of those parameters were expressed as means ± standard deviation ($n = 4$). Duncan's test was used for multiple comparisons when the differences were found to be significant ($p < 0.05$). The alpha diversity indexes of intestinal microbiota were analyzed with the QIIME software (v1.8.0, http://bio.cug.edu.cn/qiime, accessed on 28 December 2018). The relative abundance of intestinal microbiota about the genus level was tested to the Kruskal Wallis examination using R-Statistical v3.6.0 software (R Statistical Corp., Vienna, Austria) for determining bacterial differences.

## 3. Results

### 3.1. Digestive Enzyme Activities in the Intestine

As shown in Table 1, the activities of lipase and protease of the HIS group were significantly decreased in comparison with those of the control group ($p < 0.05$). The lipase activity of the H + OPC groups was significantly higher than that of the HIS group ($p < 0.05$). The protease activity of the H + OPC groups was similar to those of the control group and the HIS group ($p > 0.05$). There was no significant difference in amylase activity among all the different groups ($p > 0.05$).

**Table 1.** The digestive enzyme activities in the intestine of juvenile American eels from different groups.

| Items | Groups | | | |
|---|---|---|---|---|
| | Control | HIS | H + OPC I | H + OPC II |
| Amylase (U/mg prot) | 0.40 ± 0.01 | 0.39 ± 0.02 | 0.39 ± 0.01 | 0.43 ± 0.01 |
| Lipase (U/mg prot) | 15.35 ± 0.92 [b] | 9.08 ± 1.11 [a] | 16.97 ± 1.68 [b] | 14.08 ± 0.96 [b] |
| Protease (U/mg prot) | 182.30 ± 10.30 [b] | 127.33 ± 13.23 [a] | 158.32 ± 19.26 [ab] | 144.99 ± 18.76 [ab] |

Values are means ± SD of four replications. Different superscript letters in the same row signify significant differences ($p < 0.05$).

### 3.2. Antioxidant Potential in the Intestine

As shown in Table 2, compared with the control group, the level of MDA in the HIS group was significantly increased ($p < 0.05$), while the level of T-AOC and activity of GSH-PX and CAT were significantly decreased ($p < 0.05$). The level of MDA was significantly decreased, and the T-AOC level and activity of GSH-PX were significantly increased with OPC supplementation ($p < 0.05$). The levels of MDA and T-AOC were similar between the HIS group and H + OPC groups ($p > 0.05$). No significant difference in the SOD activity was found among different groups ($p > 0.05$).

**Table 2.** The antioxidant potential in the intestine of juvenile American eels from different groups.

| Items | Groups | | | |
|---|---|---|---|---|
| | Control | HIS | H + OPC I | H + OPC II |
| MDA (nmol/mg prot) | 2.11 ± 0.41 [a] | 2.94 ± 0.54 [b] | 2.18 ± 0.22 [a] | 1.58 ± 0.45 [a] |
| T-AOC (mmol/g prot) | 1.24 ± 0.18 [b] | 0.84 ± 0.16 [a] | 1.23 ± 0.06 [b] | 1.49 ± 0.10 [b] |
| CAT (U/mg prot) | 8.89 ± 2.01 [b] | 5.32 ± 0.81 [a] | 6.02 ± 0.51 [a] | 5.92 ± 1.46 [a] |
| SOD (U/mg prot) | 37.21 ± 2.51 | 33.33 ± 4.61 | 37.71 ± 5.27 | 34.19 ± 3.26 |
| GSH-PX (U/mg prot) | 20.89 ± 2.30 [c] | 13.14 ± 1.37 [a] | 21.23 ± 1.84 [c] | 16.80 ± 1.97 [b] |

Values are means ± SD of four replications. Different superscript letters in the same row signify significant differences ($p < 0.05$).

### 3.3. Parameters Related to Intestinal Barrier Function

As shown in Figures 1 and 2, The DAO activity and D-lac level in the control group and H + OPC groups were significantly lower than those of the HIS group ($p < 0.05$), and there were no significant differences among the other groups except for the HIS group ($p > 0.05$).

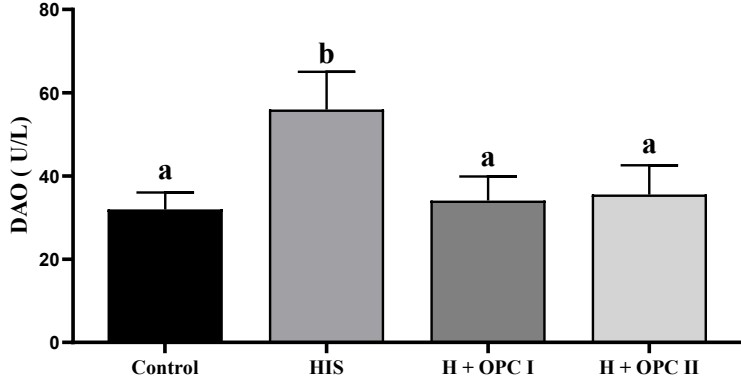

**Figure 1.** The DAO activity in the serum of juvenile American eels from different groups. Values are means ± SD of four replications. Different superscript letters on the bars signify significant differences ($p < 0.05$).

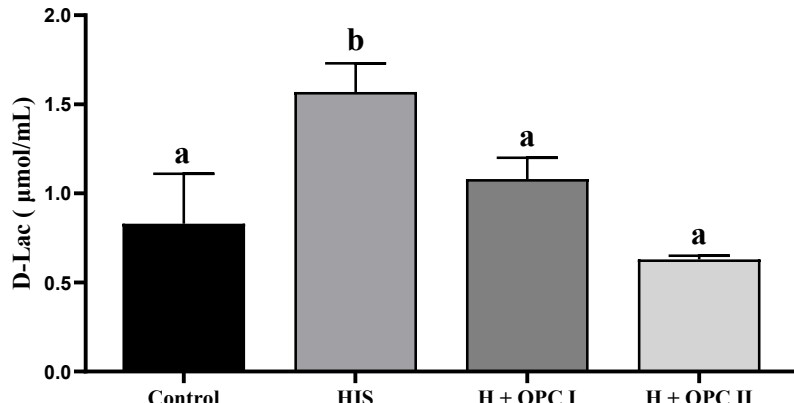

**Figure 2.** The D-lac level in the serum of juvenile American eels from different groups. Values are means ± SD of four replications. Different superscript letters on the bars signify significant differences ($p < 0.05$).

The intestinal H&E-stained sections are shown in Figures 3 and 4. The villus height of the intestine in the control group and H + OPC groups was higher than that in the HIS group. The intestinal villus height in the HIS group was significantly lower than that in the control group ($p < 0.05$). The intestinal villus height of H + OPC groups was increased to some trend in comparison with that of the control group ($p > 0.05$). The intestinal villus height of the H + OPC groups was similar to that of the control group ($p > 0.05$). There were no significant differences in the parameters of intestinal barrier function between H + OPC I and H + OPC II groups ($p > 0.05$).

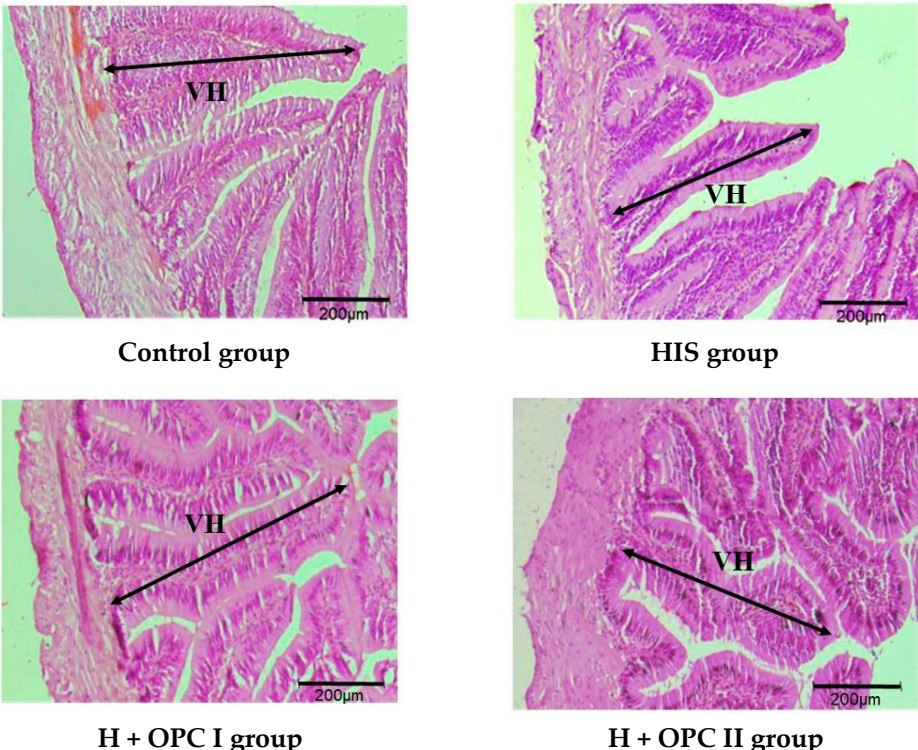

**Figure 3.** The intestinal H&E-stained sections of juvenile American eels from different groups (Magnification ×100). VH = villus height.

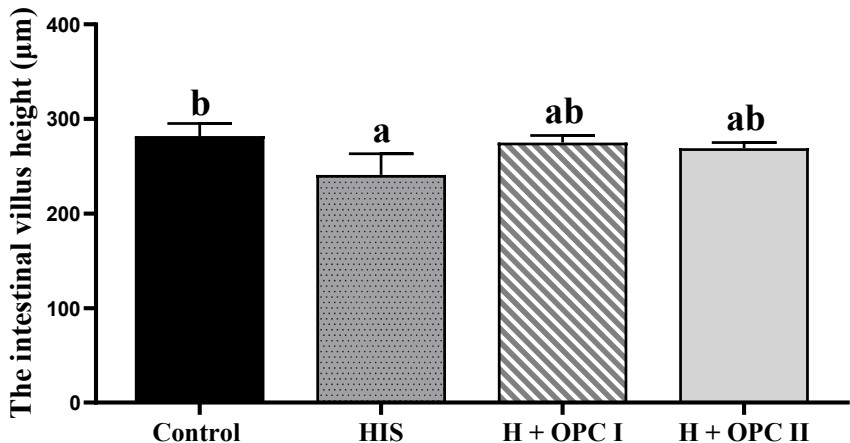

**Figure 4.** The intestinal villus height of juvenile American eels from different groups. Values are means ± SD of four replications. Different superscript letters on the bars signify significant differences ($p < 0.05$).

### 3.4. Intestinal Microbiota Analysis

### 3.4.1. Alpha Diversity of Intestinal Microbiota

As shown in Figure 5, the indexes of Chao 1, Observed_species, and PD_whole_tree in the HIS group were significantly lower than those in the other groups ($p < 0.05$), and there were no significant differences in those indexes between the H + OPC groups and control group ($p > 0.05$). The Shannon index in the H + OPC I group was significantly higher than that in the HIS group ($p < 0.05$), and there was no significant difference in the Shannon index among the control group, HIS group, and H + OPC II group ($p > 0.05$).

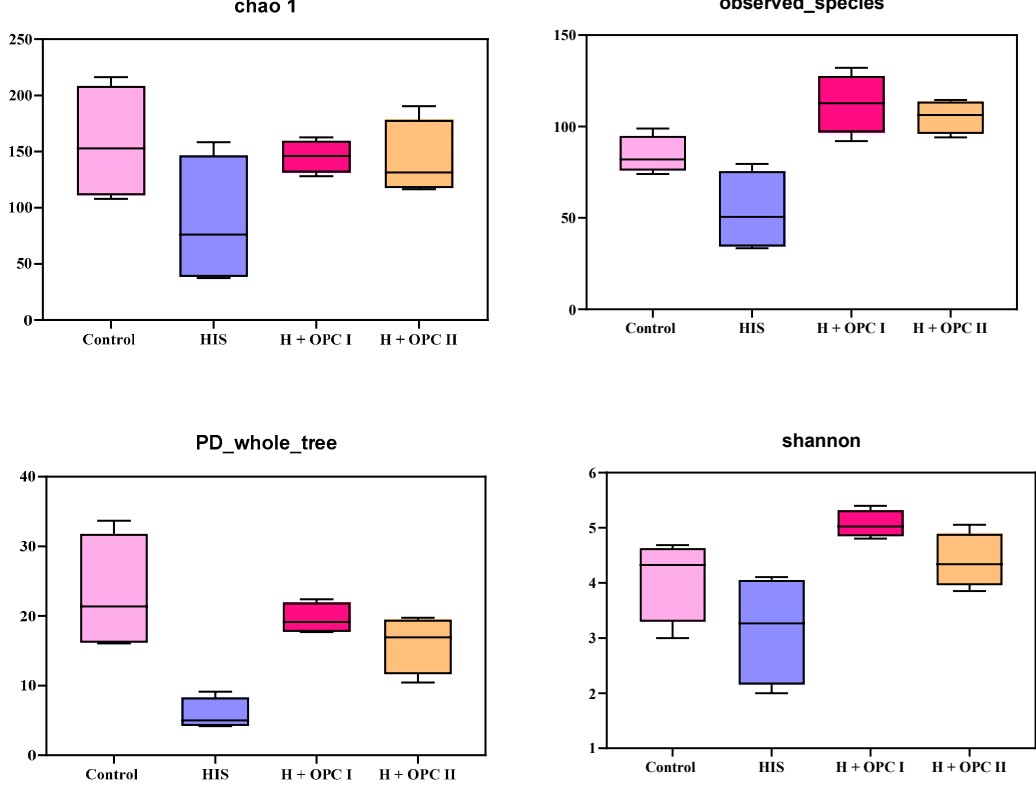

**Figure 5.** The alpha diversity indexes of intestinal bacteria of juvenile American eels in different groups.

### 3.4.2. Intestinal Microbiota at the Phylum Level

As shown in Figure 6, compared with the control group, the relative abundance of Proteobacteria in the HIS group was increased, and the relative abundance of Firmicutes was decreased. The relative abundances of Proteobacteria and Fusobacteria in the H + OPC groups were lower than those in the HIS group, and the relative abundances of Firmicutes in the H + OPC groups were higher than that in the HIS group.

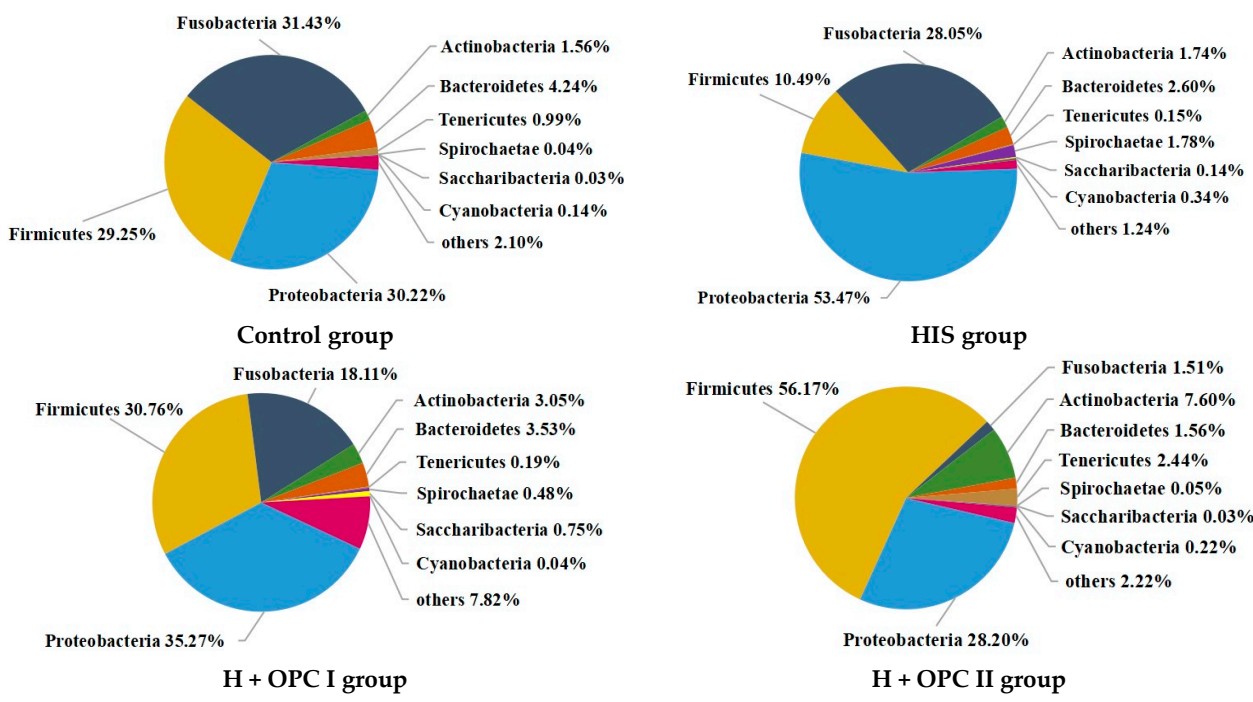

**Figure 6.** The intestinal bacteria at the phylum level of juvenile American eels in different groups.

### 3.4.3. Intestinal Microbiota at the Genus Level

As shown in Figure 7, the relative abundances of *Acinetobacter*, *Pseudomonas*, and *Aeromonas* in the HIS group were significantly increased in comparison with those of the control group and H + OPC groups ($p < 0.05$), and these bacteria of the H + OPC groups were decreased significantly in comparison with those of the HIS group and control group ($p < 0.05$). Compared with the HIS group, the relative abundance of *Lactobacillus* was significantly higher in the H + OPC groups and control group ($p < 0.05$), and the relative abundance of *Bacillus* was significantly higher in the H + OPC I group in comparison with those of the other groups ($p < 0.05$).

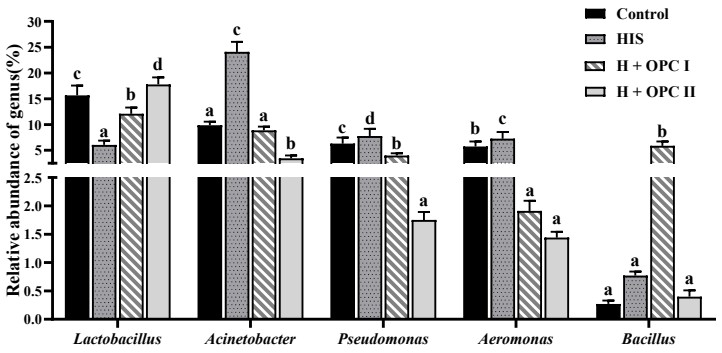

**Figure 7.** The intestinal bacteria at the genus level of juvenile American eels in different groups. Values are means ± SD of four replications. Different superscript letters on the bars signify significant differences ($p < 0.05$).

## 4. Discussion

### 4.1. Effects of OPC on the Intestinal Digestive Enzyme Activities of Juvenile American Eels Exposed to the High Level of Dietary Histamine

The activities of intestinal digestive enzymes are the indispensable parameter reflecting the capacity of absorbing and digesting in fish [21]. In this study, the intestinal lipase and protease activities were decreased by feeding with a high level of histamine diet. The previous study identified that 350 mg/kg histamine supplementation could decrease the intestinal lipase and protease activities of juvenile American eels [22]. The dietary histamine level in the HIS group was 534 mg/kg, which could certainly lower the intestinal lipase and protease activities of trial fish. The same results were also reported in the previous studies of American eels fed diets with different histamine levels [5,22]. The digestive enzyme activities in the intestine were also decreased by dietary histamine in the studies of hybrid grouper [4] and Chinese mitten crab (*Eriocheir sinensis*) [23]. Those might contribute to the damage of intestinal mucosa by dietary histamine [24]. In the current research, the lowered activities of lipase and protease in the intestine of juvenile American eels fed with a high level of dietary histamine were increased to a certain extent by OPC supplementation. A similar result was shown in a previous study of pearl gentian grouper fed with dietary 800 mg/kg OPC supplementation under dietary cadmium stress [17]. Under conditions without exogenous stress, the improvement effects of OPC on the digestive enzymes were also reported in the studies of juvenile hybrid sturgeon [12], swamp eel (*Monopterus albus*) [25], tilapia [26], and American eel [27]. The amelioration of digestive enzyme activity is probably relevant to the fact that OPC up-regulates the expression of the intestinal tight junction protein, which might improve barrier integrity and promote the secretion of digestive enzymes indirectly [28].

### 4.2. Effects of OPC on the Intestinal Antioxidant Potential of Juvenile American Eels Exposed to a High Level of Dietary Histamine

MDA is an indicator of lipid peroxidation degree [19]. T-AOC is an emblematical parameter to reflect the antioxidant capacity as a whole [29]. CAT, SOD, and GSH-PX, as essential oxidative enzymes, can scavenge excess oxygen free radicals [30]. It has been proven that the 534 mg/kg histamine in the diet could inhibit the antioxidant enzyme activities and increase the production of ROS in the intestine of juvenile American eels, which ultimately might induce oxidative stress [5]. Oxidative stress has been considered as the major trigger to induce other detrimental effects. In this study, the MDA level was increased, and the CAT and GSH-PX activities were significantly lower in the HIS group in comparison with the control group. The consistent results were also found in the studies of striped catfish (*Pangasianodon hypophthalmus*) and grouper (*Epinephelus coioides*) under dietary histamine stress [24,31]. The previous study indicated that histamine might mediate the H1R receptor to stimulate excessive ROS production and lower the activities of antioxidant enzymes, which could ultimately contribute to oxidative stress [32]. However, OPC could decrease the MDA level and increase the T-AOC level and GSH-PX activity in the intestine of trial fish in the present study, and the antioxidant status of the trial fish was similar to that of the control group. The changes in antioxidant parameters in this study suggested that OPC could combat the antioxidant potential inhibition caused by the high level of dietary histamine. Similarly, OPC eliminating the inhibition of antioxidant potential in the intestine caused by dietary cadmium stress was shown in tilapia and pearl gentian grouper [8,17]. Additionally, the antioxidant potential improvement in the intestine by OPC supplementation was considerably reported in American eel [27], rainbow trout (*Oncorhynchus mykiss*) [33], spotted sea bass (*Lateolabrax maculatus*) [34], and grass carp (*Ctenopharyngodon idella*) [35] under the condition without designed dietary stressors. There were also similar findings about the improvement of antioxidant potential by OPC supplementation in the liver of tilapia [26], goldfish (*Carassius auratus*) [36], and carp (*Cyprinus carpio* L.) [37], and in the serum of hybrid sturgeon [12] and red tilapia [38]. Those results might be attributed to OPC providing delocalized

electrons through conjugated double bonds in a poly hydroxyl structure to chelate free radicals in vivo, such as superoxide anion radical ($\cdot O_2^-$), active nitrogen (NO), hydroxyl radical ($\cdot OH$), and hydrogen peroxide ($H_2O_2$) [37]. Moreover, OPC has been found to attenuate oxidative stress by restraining the expression of nuclear factor-kappaB (NF-κB) to up-regulate the mRNA expression of certain antioxidizes [10].

### 4.3. Effects of OPC on the Intestinal Barrier Function of Juvenile American Eels Exposed to a High Level of Dietary Histamine

As the biochemical index in serum, the DAO and D-lac were used as the indicators of intestinal permeability in fish; their values might increase in the serum and reflect the damage to the intestinal mucosa barrier [39,40]. In the current study, the DAO activity and D-lac level of juvenile American eels fed a diet with high levels of dietary histamine were increased, which suggested that intestinal barrier function might be compromised with the increasing permeability in intestinal enterocytes under dietary histamine stress. These results were similar to the previous studies in American eels fed a diet with 500 mg/kg histamine [22] and hybrid grouper with exposure to a dietary histamine level being 404 mg/kg [4]. The impaired intestinal barrier function might be related to excessive histamine destructing the connection structure between intestinal mucosal cells and decreasing the relative expressions of tight junction proteins [4,24]. It was demonstrated that OPC supplementation could lower the values of the D-lac and DAO in the serum of juvenile American eel under dietary histamine stress in the present trial, which was similar to the results of spotted sea bass fed with 1 g/kg OPC under oxidized fish oil-induced stress [34]. There are few other reports about OPC protecting the intestinal barrier function in aquatic animals. In terrestrial animals, there was also similar information about OPC lowering those two parameters of broiler chicks exposed to lipopolysaccharide stress [41], weaned piglets [42], and geese [43] under the condition without exogenous stressor. The protection effects of OPC on intestinal barrier function might be relevant to its exerting antioxidant function to strengthen the intestinal tight junction protein [34], this point should be proven in future studies.

The intestinal villus height is one of the indispensable factors to directly affect the absorption capacity, and it is a typical indicator to directly evaluate the intestinal health status of fish [44]. In the current research, the intestinal villus height of juvenile American eels in the HIS group was significantly decreased in contrast to that of the control group, which indicated that dietary histamine could impair the structure of the intestine and decrease the digestive capacity of dietary nutrients. The same results were discovered in a previous study of juvenile American eels fed with dietary 350 mg/kg histamine [22]. Similar reports were verified in the yellow catfish [45,46] and Atlantic salmon [47] under dietary histamine stress. Additionally, the decrease of intestinal villus height can inhibit the digestion and absorption of dietary nutrients, which may lead to a lower growth rate of American eels. The corresponding results have confirmed that dietary histamine could significantly decrease the weight gain rate of juvenile American eels in our previous research [7]. Those might contribute to the damage of intestinal inflammation by decreasing the expression of interleukin-10 (IL-10) and increasing tumor necrosis factor-alpha (TNF-α) expression to destroy intestinal structure [45,48]. With the supplementation of OPC, the intestinal villus height of the American eels was increased significantly, which indicated that OPC could alleviate the intestinal injury caused by dietary histamine stress. To our knowledge, there were no other report about dietary OPC supplementation improving the intestinal morphology of fish under stress conditions from other sources. The promotion effects of OPC on intestinal villi were also found in some fish species, such as American eel [27], grass carp [35], and sea bass [49], bearing no stress factors. Those beneficial effects of OPC might be related to inhibit the biosynthesis of pro-inflammatory mediators, such as chemokines, adhesion molecules (CAMs), and TNF-α [28,50,51]. Further studies should be conducted to prove the detailed and reasonable mechanism about the strengthening intestinal barrier function.

*4.4. Effects of OPC on the Intestinal Microbiota of Juvenile American Eel Exposed to a High Level of Dietary Histamine*

The intestinal microbiota plays an important role on fish health [45], and its composition is easily regulated by dietary ingredients [52]. In general, alpha diversity is a well-defined parameter to evaluate the diversity and abundance of the microbiota, and the higher values of those parameters are considered beneficial to regulate intestinal health status [12,16]. The values of those parameters about alpha diversity of intestinal microbiota in the HIS group were lower than those of the control group, which indicated that the relative abundance and diversity of the intestinal microbiota might be decreased by the high level of dietary histamine. Similar results have been demonstrated in previous research about the effects of dietary histamine stress on the juvenile American eel [22], grouper [31], and yellow catfish [45]. With 300 mg/kg OPC supplemented in the diet, the lower values of parameters related to alpha diversity of intestinal microbiota were increased in present research. The improvement of the abundance and diversity of the intestinal microbiota was also found in hybrid sturgeon [12], juvenile American eel [22], and carp [53] fed diets with OPC supplementation under no designed stress condition.

Firmicutes are generally considered probiotics in the intestine, and Proteobacteria are normally considered opportunistic pathogenic bacteria [54], such as *Vibrio cholerae*, *Escherichia coli*, *Helicobacter pylori*, *Salmonella*, etc. [55]. Some bacteria of Fusobacteria might cause pro-inflammatory reactions in the host [56]. The relative abundance of Proteobacteria in the intestine was increased by the high level of dietary histamine, which indicated that American eel might have an increased probability of being infected by pathogenic bacteria. Similar changes in the composition of intestinal microbiota were observed in some studies of American eel [22], striped catfish [24], grouper [31], and yellow catfish [45] fed diets with high levels of dietary histamine. From the results in the intestinal microbiota composition of American eels in H + OPC groups, it was observed that the increased relative abundance of Firmicutes was accompanied by the decreased relative abundances of Proteobacteria and Fusobacteria in comparison with those of the HIS group. Those results indicated that OPC could counteract the detrimental effects of dietary histamine stress on the composition of the intestinal microbiota of American eels by altering the ratio of potential probiotics and pathogens at the phylum level.

In the present study, *Acinetobacter*, *Pseudomonas*, *Aeromonas*, *Lactobacillus*, and *Bacillus* in the intestine of juvenile American eel were significantly affected among different treatment groups. *Acinetobacter*, *Pseudomonas*, and *Aeromonas* are considered opportunistic pathogens in aquaculture [24]. *Lactobacillus* and *Bacillus* could act as probiotics to improve intestinal health [57,58]. For the bacteria at the genus level, the increased relative abundances of *Acinetobacter*, *Pseudomonas*, and *Aeromonas* were accompanied by the decreased relative abundances of *Lactobacillus* and *Bacillus* in the HIS group, which were similar to the previous research of American eels fed with 650 mg/kg histamine in the diet [22]. Consistent results about the changes in relative abundances of potential probiotics and pathogenic bacteria at the genus level caused by high levels of dietary histamine were also reported in striped catfish [24], grouper [31], and yellow catfish [45].

The increased relative abundances of *Bacillus* and *Lactobacillus* were accompanied by the decreased relative abundances of *Acinetobacter*, *Pseudomonas*, and *Aeromonas* in the intestine of juvenile American eel from H + OPC groups, which indicated that OPC could regulate the species and population of intestinal microbiota beneficially. At present, few studies have reported the effects of OPC supplementation on the intestinal microbiota of different fish species under various types of stressors. However, the positive effects of OPC regulating the intestinal microbiota were confirmed in several studies conducted without the designed dietary stress. It was found that 400 mg/kg OPC could increase the relative abundance of intestinal probiotics of juvenile American eel [16]. Dietary 50 mg/kg OPC supplementation could decrease the relative abundance of *Aeromonas* in the intestine of hybrid sturgeon [12]. OPC might act as the substrate to stimulate probiotic colonization in the intestine of carp [53]. In a *vitro* experiment, OPC was found to alleviate

the amount of *Escherichia coli* and *Salmonella typhimurium* adhered to IPEC-J2 cells [59]. The above results indicated that the OPC could beneficially regulate the intestinal microbiota. Previous studies have demonstrated that the antibacterial capacity of OPC might be mainly manifested to destroy the integrity of the cellular structure by combining the hydroxyl in OPC's chemical structure with the lipid bilayer of the pathogenic bacteria cell membrane, thus inhibit the proliferation of the opportunistic pathogens [28,60].

## 5. Conclusions

In conclusion, dietary 300 mg/kg oligomeric proanthocyanidins supplementation could alleviate the detrimental effects of dietary 534 mg/kg histamine on the intestinal health of juvenile American eels by improving the digestive enzymes activities, antioxidant potential, and intestinal barrier functions and regulating intestinal microbiota beneficially. There was no further improvement effect of dietary 600 mg/kg oligomeric proanthocyanidins on the growth performance and intestinal health of juvenile American eels. The optimal level of dietary OPC supplementation to counteract the stress from different dietary histamine levels should be confirmed in further study.

**Author Contributions:** Methodology, Y.H. and S.Z.; validation, Y.H., S.W. and S.Z.; formal analysis, Y.H. and S.W.; investigation, Y.H., S.W. and S.Z.; data curation, F.X., Y.L. and S.Z.; writing—original draft preparation, S.W.; writing—review and editing, S.W. and S.Z.; project administration, S.Z.; funding acquisition, S.Z. All authors have read and agreed to the published version of the manuscript.

**Funding:** This study was supported by the earmarked fund for the China Agriculture Research System of MOF and MARA (CARS-46).

**Institutional Review Board Statement:** The study was conducted according to the guidelines of the Declaration of Helsinki and was approved by the Animal Care Advisory Committee of Jimei University (Approval No. 2018-0912-001, 12 September 2018).

**Informed Consent Statement:** Not applicable.

**Data Availability Statement:** The data used during the current study are available from the corresponding author upon reasonable request.

**Acknowledgments:** The authors thank Keding Xia for providing the trial fish.

**Conflicts of Interest:** The authors declare no conflict of interest.

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
