# Peer review of "Oligomeric Proanthocyanidins Alleviate the Detrimental Effects of Dietary Histamine on Intestinal Health of Juvenile American Eels (Anguilla rostrata)"

_fishes, doi:10.3390/fishes8080413_

Round 1
Reviewer 1 Report
Wang S. et al. have submitted a paper titled “Oligomeric Proanthocyanidins Alleviate the Negative Effects of Intestinal Health Caused by Dietary Histamine in Juvenile American Eels (Anguilla rostrata)”. Although the manuscript is interesting, there are no molecular mechanism data. Most of the data are only observations between groups. Also, no group was fed only OPCs. These limitations make this manuscript weak.
Major comments
1. I wonder why the author analyzed only 4 eels per group. There are very few of them. The authors should increase the number for an accurate analysis, taking into account individual differences in eels.
2. Current data show that the HIS group had a reduced level of intestinal absorption due to damaged intestinal mucosa and shortened villi. These factors can affect the growth rate of eels. The conclusion would be much clearer if the author provided basic information such as the weight and length of the eel.
3. Add the detail description how to compare with in the legend.
4. In order to clear their conclusion, authors should provide intestinal inflammation data such as mRNA or protein level.
Author Response
Wang S. et al. have submitted a paper titled “Oligomeric Proanthocyanidins Alleviate the Negative Effects of Intestinal Health Caused by Dietary Histamine in Juvenile American Eels (Anguilla rostrata)”. Although the manuscript is interesting, there are no molecular mechanism data. Most of the data are only observations between groups. Also, no group was fed only OPCs. These limitations make this manuscript weak.
Re:many thanks for your suggestions, which would help us in the future study about the molecular mechanism of oligomeric proanthocyanidins alleviating the histamine stress in American eels.
Major comments
I wonder why the author analyzed only 4 eels per group. There are very few of them. The authors should increase the number for an accurate analysis, taking into account individual differences in eels.
Re: Thank you for your careful check. However, the description of “n=4” in our manuscript is the data analyzed of four replications per group (there were four tanks in each group), which does not mean that there are only four eels to be analyzed. The number of samples utilized for analysis from each tank has been described in the part of “2.2 Sample Collection”.
Current data show that the HIS group had a reduced level of intestinal absorption due to damaged intestinal mucosa and shortened villi. These factors can affect the growth rate of eels. The conclusion would be much clearer if the author provided basic information such as the weight and length of the eel.
Re: Thank you for your suggestion. The results of the growth performance of the present feeding trial have been published in a previous article (Zhai et al., 2020). The related information such as the weight and length of the eels is not suitable to publish again, but we discussed the relation between the damaged intestinal mucosa and shortened villi and the related growth performance in the revised manuscript.
Zhai, S.W.; Wang, Y.; He, Y.X.; Chen, X.H. Oligomeric proanthocyanidins counteracts the negative effects of high level of dietary histamine on American eel (Anguilla rostrata). Front. Mar. Sci. 2020, 7, 549145, doi: 10.3389/fmars.2020.549145.
Add the detail description how to compare with in the legend.
Re: Thank you for your suggestion. The descriptions have been added in the legend. Please see the revised manuscript.
In order to clear their conclusion, authors should provide intestinal inflammation data such as mRNA or protein level.
Re: We sincerely appreciate your valuable suggestion, and we are confident that it would be useful to clear conclusion by providing the intestinal inflammation data such as mRNA or protein level. Unfortunately, we did not conduct the determination of those parameters in the present trial because of the insufficient intestinal samples. We might measure those parameters in future research.
Besides, we made extensive modifications to lower the repetition rate of present manuscript, the revised places in our manuscript were highlighted in red color.
Reviewer 2 Report
The study appears to have been conducted appropriately and the results accurately support the conclusions, however I have a few issues with the presentation of methods. The diet formulation and manufacturing section appears to cite other work by this same group and following up on that citation is a paper that is almost identical to this one but utilizing some metabolomics analyses. The conclusions of that paper are very similar to the conclusions of this paper, they have just measured a few additional parameters and arrived at the same conclusion about 300 mg/kg OPC. Also, why such a variation in the number of animals utilized for each analysis from each tank?
Parts of the manuscript are very difficult to decipher. Suggest having a native English speaker proofread the manuscript for grammar, sentence structure, and overall clarity.
Author Response
The study appears to have been conducted appropriately and the results accurately support the conclusions, however I have a few issues with the presentation of methods. The diet formulation and manufacturing section appears to cite other work by this same group and following up on that citation is a paper that is almost identical to this one but utilizing some metabolomics analyses. The conclusions of that paper are very similar to the conclusions of this paper, they have just measured a few additional parameters and arrived at the same conclusion about 300 mg/kg OPC. Also, why such a variation in the number of animals utilized for each analysis from each tank?
Re: It was true that the present manuscript and the previous published article were all from one feeding trial, and different parameters were measured, respectively. The previous published article focused on the parameters of growth, serum biochemistry (without DAO and D-lac), and liver metabolites, while the present manuscript focused on the parameters related to intestinal health, and they arrived at the same conclusion about different OPC supplementation to alleviate the negative effects of dietary histamine on the corresponding parameters.
In our experiment, it was very difficult to collect blood samples from the tail vein of the juvenile American eel. They were too small. Therefore, more eels were needed to collect sufficient blood samples for analysis of serum parameters, especially the DAO and D-lac. Besides, the detailed number of intestinal samples for different parameters in the present manuscript was clearly shown in the revised manuscript, please see the revised manuscript.
Parts of the manuscript are very difficult to decipher. Suggest having a native English speaker proofread the manuscript for grammar, sentence structure, and overall clarity.
Re: Thanks for your suggestion. We have tried our best to polish the language in the revised manuscript. And we hope it will satisfy the requirements of Fishes.
Besides, we made extensive modifications to lower the repetition rate of the present manuscript, the revised places in our manuscript were highlighted in red color.
Reviewer 3 Report
General comments:
The ms investigates the effects of dietary Oligomeric Proanthocyanidins (OPC) supplementation on the intestinal health of juvenile American eels exposed to a high histamine diet. The main contribution of this study is that it provides significant evidence that OPC supplementation can mitigate negative effects of high histamine levels by improving digestive enzyme activity, antioxidant potential, and intestinal barrier functions, and by beneficially regulating intestinal microbiota. The strength of this study lies in its comprehensive analysis approach, which includes digestive enzyme activity, antioxidant parameters, intestinal barrier function assessments, and microbiota analysis.
The study is well-conceived and fills a significant gap in our understanding of the effects of OPC supplementation on fish health under high histamine dietary conditions. The hypothesis that OPC supplementation can mitigate the effects of a high histamine diet in American eels is effectively supported by the experimental design and results.
The study could benefit from a clearer organization, particularly in the discussion section which is somewhat lengthy and packed with information. This could make it difficult for readers to follow the logical flow of the argument. I suggest the authors consider enhancing the structure of the discussion section. Specifically, it would be advantageous to delineate the discussion of different results into distinct paragraphs or subsections. This could include separate sections for digestive enzymes, antioxidant parameters, and intestinal barrier function, among others. Such a structured approach would substantially improve the readability and comprehensibility of the ms.
There are a few minor grammatical errors and awkward phrases that should be revised for clarity and correctness.
Author Response
The study could benefit from a clearer organization, particularly in the discussion section which is somewhat lengthy and packed with information. This could make it difficult for readers to follow the logical flow of the argument. I suggest the authors consider enhancing the structure of the discussion section. Specifically, it would be advantageous to delineate the discussion of different results into distinct paragraphs or subsections. This could include separate sections for digestive enzymes, antioxidant parameters, and intestinal barrier function, among others. Such a structured approach would substantially improve the readability and comprehensibility of the MS.
Re: Many thanks for your good suggestion. We added the subtitles in the discussion section, it is easy for readers to follow the logical flow of the discussion section and these modifications would substantially improve the readability and comprehensibility of the manuscript.
Besides, we made extensive modifications to lower the repetition rate of the present manuscript, the revised places in our manuscript were highlighted in red color.